# Recent Advances in the Discovery of Nicotinic Acetylcholine Receptor Allosteric Modulators

**DOI:** 10.3390/molecules28031270

**Published:** 2023-01-28

**Authors:** Dina Manetti, Silvia Dei, Hugo R. Arias, Laura Braconi, Alessio Gabellini, Elisabetta Teodori, Maria Novella Romanelli

**Affiliations:** 1NEUROFARBA Department, Section of Pharmaceutical and Nutraceutical Sciences, University of Florence, 50019 Florence, Italy; 2Department of Pharmacology and Physiology, College of Osteopathic Medicine, Oklahoma State University, Tahlequah, OK 74464, USA

**Keywords:** positive allosteric modulator, negative allosteric modulator, desensitization, α7, α4β2, nicotinic acetylcholine receptor

## Abstract

Positive allosteric modulators (PAMs), negative allosteric modulators (NAMs), silent agonists, allosteric activating PAMs and neutral or silent allosteric modulators are compounds capable of modulating the nicotinic receptor by interacting at allosteric modulatory sites distinct from the orthosteric sites. This survey is focused on the compounds that have been shown or have been designed to interact with nicotinic receptors as allosteric modulators of different subtypes, mainly α7 and α4β2. Minimal chemical changes can cause a different pharmacological profile, which can then lead to the design of selective modulators. Experimental evidence supports the use of allosteric modulators as therapeutic tools for neurological and non-neurological conditions.

## 1. Introduction

Nicotinic acetylcholine receptors (nAChRs) belong to the family of Cys-loop Ligand-Gated Ion channels (LGIC), composed of five subunits assembled around a central pore. The different combinations of the 17 cloned subunits give a high number of possible receptor subtypes, expressed in the Central (CNS) and Peripheral (PNS) Nervous System as well as in non-neuronal cells and other peripheral organs. The most abundant receptors in the CNS are the homomeric α7 and heteromeric α4β2 subtypes [1,2]. The widespread distribution of nAChRs and their involvement in physiological and pathological processes make such receptors important drug targets for different therapeutic applications.

The first generation of drugs acting on nAChRs were compounds presumed to bind to the same site as that for the endogenous neurotransmitter acetylcholine (ACh) (so called “orthosteric sites”), including agonists and competitive antagonists. Agonists bind to orthosteric sites, activating the receptor to a greater or a lesser extent depending on their efficacy and intrinsic potency. More recently, however, attention has been directed towards ligands interacting with binding sites located apart from orthosteric sites (so called “allosteric sites”). Allosteric modulators do not directly activate/inactivate the receptor but indirectly regulate (positively or negatively) the activity of ACh and other agonists. Only a few modulators have the capacity to indirectly activate nAChRs (so called “allosteric agonists”) [3]. 

Positive allosteric modulators (PAMs) have several advantages over orthosteric ligands, the most important being that the receptor activation does not exceed the physiological stimulus produced by the endogenous neurotransmitter. For example, type II PAMs reduce the likelihood of agonist-induced α7 nAChR desensitization, thus providing a tool for escaping tolerance and overdose, but opening the possibility for Ca^2+^-induced cytotoxicity [4]. Since type I PAMs do not delay the process of α7 nAChR desensitization, they would not induce cell toxicity as in the case of type II PAMs (vide infra). PAMs do not upregulate nAChRs as in the case with agonists and even competitive antagonists [5]. 

Another advantage is related to the problematic aspect of drug discovery, namely the selectivity of ligands towards the different receptor subtypes. In the case of nAChRs, the orthosteric site is highly conserved providing very little space for the design of selective agents for the various nAChR subtypes. On the contrary, allosteric sites are often less conserved, could be located anywhere on the receptor surface, and therefore could be specific for a particular receptor subtype. All these factors could therefore provide ligands with exquisite receptor selectivity [6]. The limited clinical efficacy observed for orthosteric ligands could be due to overdosing, receptor desensitization or upregulation and poor selectivity towards the various receptor subtypes. The discovery of drugs has therefore moved towards the allosteric modulation of the receptor which, as previously mentioned, gives a more physiological response that reduces or cancels the drawbacks that instead occur with orthosteric ligands. 

Orthosteric agonists activate and subsequently desensitize the receptor. Several models have been proposed to illustrate the multiple conformations of the nAChR and the transitions from one state to another in the presence of ACh or another agonist. The four-state model proposed by Williams et al. comprises the closed, open, intermediate, and inactive desensitized state of the nAChR ion channel [7]. Molecular events associated with the nAChR function are initiated by binding of ACh. Agonist-evoked activation induces long-range conformational changes in the receptor structure (so called “gating”) [8] that permits the opening of the intrinsic cation channel, allowing the influx of Na^+^ and Ca^2+^ and efflux of K^+^, and consequently, the membrane depolarization. The rapid activation of the receptor from the closed to open state occurs in the μs to ms time regime. In the resting state, the receptor has low affinity for the agonist, and consequently a very low likelihood for the bound form, and thus the ion channel is essentially closed. The resting state constitutes approximately 80% of the receptor population while the remaining 20% exists in the desensitized state [7]. In the prolonged presence or high concentrations of the agonist, the receptor passes from an intermediate transition state to a desensitized state in the ms time regime [8]. Agonist-induced nAChR desensitization occurs rapidly, and among nAChR subtypes, α7 nAChR desensitization is the fastest. The rate of desensitization for β2 containing subtypes is faster than that for β4 containing subtypes [9]. The N-terminal region of the β subunit has been shown to confer the different rate of desensitization of the β-containing heteromeric nAChRs [10]. The desensitized state is a different closed conformation, where the receptor has high affinity for the neurotransmitter, but the channel is in a non-conducting form. Desensitization of nAChRs is modulated by protein kinases and phosphatases, which, respectively, phosphorylate and dephosphorylate the receptor [11]. 

In this review, we focused on synthetic and natural nicotinic allosteric modulators reported in the literature from 2010 to 2022 acting on various nAChRs. The search was performed through Pubmed using the following keywords: α7 modulators, α4β2 modulators, nicotinic allosteric modulators, and natural nicotinic allosteric modulators; the search predominantly retrieved α7 receptor ligands. Therefore, after a brief overview on α7 receptors and the various mechanisms of its allosteric modulation, the different classes of α7 PAMs have been described, followed by those acting on α4β2 and other nAChR sybtypes, and ending with the negative allosteric modulators of all subtypes. The review aims to provide a stimulus for reflection on the importance of the allosteric modulation of the nicotinic receptors.

## 2. α7 Nicotinic Acetylcholine Receptors

α7 nAChRs are abundantly expressed in several brain areas, including the prefrontal cortex, hippocampus, and other limbic structures such as the amygdala, hypothalamus, and nucleus accumbens [12]. An approach based on the positive modulation of α7 nAChR (i.e., potentiation) could be useful for the treatment of various CNS disorders such as cognitive impairment, acute and chronic neurodegenerative conditions, different pain syndromes, schizophrenia, depression, smoking cessation, and cough suppression [2,13].

The α7 nAChR subtype shows important characteristics such as rapid desensitization upon activation as well as selective inhibition by α-bungarotoxin and methyllycaconitine. Another important feature of α7 nAChRs is the relatively higher permeability to Ca^2+^ compared to other nAChR subtypes. This gives the α7 nAChR a role not only in the generation of electrical signal in the synapses, but also in the direct activation of Ca^2+^-dependent intracellular processes, including neurotransmitter release, neuroprotection and synaptic plasticity [14]. Furthermore, choline, a byproduct of ACh hydrolysis by different cholinesterases, is also an endogenous agonist of α7 nAChRs [15].

The homopentameric α7 nAChR is made up of five identical subunits, which gives five potential binding sites for ACh at each α7(+)/α7(−) interface [16]. The sign (+) indicates the principal component of the subunit interface formed by loops A–C, and (−) indicates the complementary component comprised by loops D–F. There is also experimental evidence supporting the existence of endogenous heteromeric α7β2 nAChRs, but the role of this subtype is less clear [17]. α7 nAChRs can be activated when a single site is occupied, while the cooperative effect of a second bound ACh can be negative, promoting the desensitization process [18]. In this regard, some hypotheses have proposed the presence of non-equivalent binding sites, or that the desensitization of the channel is proportional to its occupancy by the ligand and that therefore a lower occupancy of the receptor leads to a slower desensitization [19,20]. In reality, the short openings of the α7 nAChR, a feature of this subtype, can be achieved with the binding of a single agonist molecule [21]. These short-lived channel openings could be a protective mechanism against Ca^2+^-mediated toxicity. Furthermore, high receptor occupancy does not seem to be required for full α7 nAChR activation, but the high number of binding sites could increase the agonist sensitivity when its concentration is low [18]. 

Agonist-induced α7 nAChR activation is followed by rapid desensitization (<100 ms) [22]. This rapid inactivation has often been described negatively. Currently, this phenomenon is considered as a sophisticated system that has evolved to control the excess of cholinergic signaling, preventing cytotoxicity leading to cell death, and to modulate synaptic plasticity. α7 nAChR desensitization is associated with structural changes in both the orthosteric site and ion pore [9]. In fact, it has been shown that the expression of hyperfunctional α7 nAChRs (due to the M2-L247T mutation) causes an abnormal increase in apoptosis and toxicity in neuronal cells [23,24,25]. This mutation has been shown to destabilize the conformation of the desensitized receptor making it more conductive to channel opening compared to the wild-type α7 nAChR.

Desensitization can occur due to continuous or repeated exposure to an agonist which results in a progressive decrease in the response to a subsequent stimulus. α7 nAChRs desensitize after exposure to high agonist concentrations, and the resulting non-conducting state of the receptor is readily reversible. The transition from the desensitized state to the resting state occurs without the need for reactivation of the receptor so that the receptor can change its conformation from the desensitized to the resting state directly. 

## 3. The Positive Allosteric Modulation of α7 nAChRs

The problems rising from the lack of selectivity of orthosteric ligands and from the ability of agonists to induce receptor desensitization and upregulation has posed a considerable challenge for the development of agents capable of modulating nicotinic cholinergic transmission through allosteric mechanisms. Such modulation can be achieved using ligands with different pharmacological and structural properties. 

In general, PAMs bind to sites different from orthosteric sites enhancing agonist-induced receptor activation. PAMs mainly have three advantages over agonists. Since typical PAMs do not act by themselves, they require the co-application of an agonist to increase the likelihood of ion channel opening induced by endogenous or exogenous agonists. Allosteric binding sites are less conserved than orthosteric sites, consequently, the former have higher structural diversity that can be exploited to obtain greater selectivity. Considering that the action of agonists determines receptor desensitization, the use of PAMs that delay receptor desensitization (e.g., type II α7-PAMs) can be extremely useful to overcome or reduce tolerance and consequently, overdose [26]. PAMs maintain the temporal characteristics of endogenous activation, reduce tolerance due to desensitization, and show higher receptor selectivity than agonists. It is probable that these positive qualities are related to the observed neuroprotection elicited by PAMs [27].

PAMs act for a limited time, amplifying the signaling mediated by endogenous cholinergic agonists (i.e., ACh and choline) giving a tone that is as physiological as possible. They increase the frequency and average duration of the α7 nAChR channel openings and the number of activations. Furthermore, type II PAMs delay receptor desensitization, reducing the energy barrier. These PAMs can also result in destabilization of the desensitized state of α7 nAChRs, allowing rapid restoration of ion channels in active conformations [28]. All these pharmacological properties make α7-PAMs ideal candidates for therapeutic uses [29].

Based on their different effects on the desensitization process, PAMs targeting α7 nAChRs can be divided into three main classes: type I (e.g., NS-1738 and LY-2087101), type II (e.g., PNU-120596, TBS-516, and PAM-2), and type I/type II intermediate modulators (e.g., SB-206553 and JNJ-1930942) (Figure 1). Type I modulators increase agonist-induced receptor activation without affecting desensitization, and type II modulators increase agonist-induced receptor activation, delay the process of desensitization, and reactivate agonist-induced desensitized receptors [30,31]. Intermediate modulators delay less efficiently the process of desensitization, having a pharmacological profile between type I and type II PAMs [32,33,34]. Figure 1 shows several examples of these different types of modulators. Changes in temperature (22 or 34 °C) have different effects between type I and type II PAMs. In fact, an increase in the desensitization rate with increasing temperature has been highlighted. Therefore, if the increase in temperature acts on desensitization, this effect will be evident above all for the modulation mediated by type II PAMs [35,36]. In this regard, PAM-2 has a temperature-sensitivite intermediate between type I and type II PAMs.

## 4. Allosteric Activating PAMs

The compounds called allosteric activating PAMs (i.e., ago-PAMs) have intrinsic agonistic activity in the absence of an agonist, through interaction with an allosteric site(s) in the N-terminal extracellular domain (ECD). Ago-PAMs such as 4BP-TQS, (+)-GAT 107, and B-973 exhibit this particular pharmacological behavior (Figure 2). Since the dose–response curve for 4BP-TQS is faster and greater than those observed with agonists, it is believed that receptor activation is achieved with minimal desensitization through a different allosteric site [3]. It has been shown that 4BP-TQS was able to allosterically activate only the α7 receptor, while it behaves as an antagonist at α3β4, α4β2, and muscle-type receptors [37]. The direct allosteric activation of ago-PAMs is mediated by interactions at the archetypical transmembrane domain (TMD) site for type II PAMs and a novel direct allosteric activation (DAA) binding site located in ECD of the receptor distinct from both the orthosteric agonist and PAM sites [16,38,39]. B-973 is an α7 PAM which slows down ACh-induced receptor desensitization and noticeably increases ACh apparent potency, supporting a type II PAM pharmacological profile. However, B-973 behaves as an ago-PAM at high concentrations, thus having a dual concentration-dependent pharmacological behavior [40].

## 5. Silent Allosteric Modulators

Neutral allosteric ligands (NALs) or silent allosteric modulators (SAMs) block the efficacy or potency of PAMs as they bind to the same allosteric sites, without affecting the response to agonists [41,42]. In other words, SAMs may act as antagonists of PAMs [42]. For example, some derivatives with general formula **2** can behave as silent allosteric modulators.

## 6. Silent Agonists

Silent agonists or “silent desensitizers” are very weak partial agonists that bind to the orthosteric site, and are capable of inducing a stable and prolonged receptor desensitization which is sufficient to block the action of other agonists [43,44]. Silent agonists-induced desensitization can be achieved without prior receptor activation, and PAMs can destabilize this desensitization by restoring the receptor in an open state. However, silent agonists are not to be confused with competitive antagonists, which are able to block the interaction of agonists with the receptor through a mechanism insensitive to PAMs. For example, NS-6740 (Figure 2) is an α7 silent agonist [45] and the co-application of this compound and a type II PAM is in fact able to restore receptor activity [46,47,48].

## 7. Allosteric Modulatory Sites at α7 nAChRs

Allosteric modulators bind to sites topographically distinct from the orthosteric sites, and are able to modulate either positively (PAMs) or negatively (so called negative allosteric modulators or NAMs) the transition states of the receptor triggered by an agonist (i.e., gating). Potential allosteric sites for PAMs have been identified in different nAChR subtypes. For example, type I α7-PAMs bind to overlapping sites at the ECD, whereas type II α7-PAMs bind to an α7/α7 intersubunit site located in the middle of the TMD comprising the M2/M3 segments [49,50]; PAM-2 (Figure 1), another type II α7-PAMs, seems to also interact with an α7/α7 intersubunit site and with the ECD-TMD junction [31].

Ca^2+^ ions were the first PAMs identified at α7 nAChRs. The Ca^2+^-binding sites were recognized through site-directed mutagenesis studies and located in particular in the ECD clustered near the interface with the TMD [51]. Other divalent cations, such as Ba^2+^ and Sr^2+^, also positively modulate α7 nAChRs interacting with the same cation modulatory site in the α7 ECD [52,53].

## 8. α7 PAMs

The chemical structure of PAMs is quite heterogeneous, and only some pharmacophore properties were elucidated. Some PAM derivatives were obtained from GABA_A_ PAM libraries, based on the homology between α7 nACh and GABA_A_ receptors [4], which belong to the same superfamily of LGICs. Structural modifications have made it possible to obtain α7 selective derivatives such as AVL-3288 [4]. Another approach, particularly exploited by medicinal chemists is the high-throughput compound screening technique, which made it possible to obtain most of the PAMs that will be analyzed in this review, confirming the structural heterogeneity of allosteric modulators. The screening of libraries of small natural molecules has also allowed to obtain PAMs for different nAChR subtypes (vide infra).

The first allosteric modulator reported in the literature is 5-hydroxyindole (5-HI) (Figure 3), a type I PAM that significantly increases ACh evoked currents with potency EC_50_ 0.63 μM, but minimal effects on the rapid desensitization of the hα7 receptor. Although 5-HI is selective for the α7 receptor with respect to other nAChR subtypes, it also modulates the 5-HT_3_ receptor [54]. 

Another potent type I PAM is NS-1738 (Figure 4), with good selectivity towards other nAChR subtypes. NS-1738, considered the archetype of type I PAMs, has pro-cognitive properties in vivo, even showing a modest brain:plasma ratio of 0.50 in rat and, consequently, a relatively modest brain penetration [55]. PNU-120596 is the first potent α7-PAM (EC_50_ = 0.16 μM at the hα7) that induces a drastic reduction in receptor desensitization. In this regard, this compound is considered the archetype of type II α7-PAMs.

PNU-120596 exhibits the same ureidic function as in NS-1738 (EC_50_ = 3.4 μM at the hα7), but with differences in one of the two aromatic rings (Figure 4). In particular, the substituted phenyl in NS-1738 is transformed in an isoxazole ring which leads to particularly evident pharmacological differences. 

PNU-120596 is among the best studied PAM in preclinical models of pain, ischemia, schizophrenia, and cognitive deficits as well as in clinical trials for the treatment of Alzheimer’s disease and schizophrenia [56]. Unfortunately, this compound has not been approved by the FDA. An important reason is that this PAM can induce cellular toxicity and death [57].

PNU-120596 binds into the TMD interface of two adjacent α7/α7 subunits. The cryogenic electron microscopy (cryo-EM) structure of the human α7 nAChR in different states and the molecular details of the PNU-bound interface have been recently described [58]. The different pharmacological profile of type I or II PAMs depends on the site to which they bind. Minimal chemical differences such as those highlighted between NS-1738 and PNU-120596, result in extremely different modulatory activity as well as distinct binding sites (Figure 5).

Numerous type I PAMs have been described in recent years obtained, as mentioned before, via high-throughput compound screening. Compounds such as LY-2087101, LY-2087133, and LY-1078733 (Figure 6) have low selectivity for α7, being also PAMs at the α2β4, α4β2, and α4β4 subtypes [59]. The modulation of both α7 and non-α7 receptor subtypes expressed centrally can be useful for the study of pathologies in which these receptors are involved, including depression, anxiety, addiction, chronic pain, and cognitive impairment, to name the most notorious and consequently in the development of novel medications. 

RO5126946, discovered and characterized by Roche through screening of arylsulfonamide small-libraries, is pharmacologically classified as a type II PAM. RO5126946 increases the peak amplitude of agonist-evoked current (EC_50_ = 0.06 μM at the hα7) and delays receptor desensitization. Unlike type II PAMs, however, it is unable to reactivate desensitized α7 receptors [60]. The derivative BNC375, with the same arylsufonamide function as in RO5126946 but with an alkylamino chain instead of an amide, has been classified as type I PAM or type II PAM depending on the stereochemistry around the central cyclopropyl ring (Figure 7). BNC375 and its analogs with stereochemistry (*R*,*R*) are identified as type I PAMs, while compounds of the same series with stereochemistry (*S,S*) are identified as type II PAMs. Furthermore, (*R*,*R*) BNC375 has a much lower peak current enhancement than its (*S*,*S*) enantiomer, increasing ACh signal without modifying receptor desensitization. Modifications made on the *ortho*-substituents, in particular with electron-withdrawing groups, lead to derivatives with reduced type II PAM characteristics, while electron-donating substituents have a stronger effect in delaying desensitization, indicating that the electron density of the aniline group plays a crucial role in the modulation of desensitization kinetic and therefore in the type I vs. type II PAM profile [61]. 

The amide function is also present in another series of compounds, including PAM-2 (EC_50_ = 4.6 μM at the hα7) and PAM-4 (EC_50_ = 5.5 μM at the hα7) (Figure 8), which enhance agonist-evoked Ca^2+^ influx in the hα7 with potencies higher than the inhibitory activity at other non-α7 nAChRs [62]. These α7-PAMs were able to reactivate desensitized receptors with EC_50_ values similar to that for potentiation, supporting a type II PAM profile [36]. The important pharmacophoric elements for PAM-4 activity are the furyl ring, where this moiety is arranged as π–π stacking with Tyr232 and the oxygen forms a H-bond with the same residue at one α7 subunit, and the amide function, where the carbonyl forms a H-bond with Asp236 on the adjacent α7 subunit [Arias, manuscript in preparation]. PAM-2 shows pro-cognitive, promnesic, antidepressant-like and anxiolitic-like activities in mice [63,64,65,66].

PAM-4, PAM-2 and its analog DM489 are able to reduce neuropathic pain, allodynia and inflammation in mice using different animal models [67,68].

To assess whether there is a direct correlation between the observed therapeutic effects and brain penetration, the blood–brain barrier permeability (i.e., LogBB) of several compounds was calculated [67,68]. A high brain permeability will allow the molecule to reach the brain target. The calculated LogBB values for PAM-2 (0.09), DM489 (0.17) and PAM-4 (−0.06) indicate that PAM-2 and DM489 not only have higher brain penetration than PAM-4, but also might induce more prolonged effects due to a potential higher accumulation in fatty tissues.

Other selective type II α7-PAMs with an arylsulfonamide ring are shown in Figure 9. Both TQS and A-867744 cause little or no activation of recombinant and native α7 nAChRs in the absence of an agonist. TQS, a naphtalenyl-cyclopentaquinoline derivative [69], has a potency (EC_50_ = 0.2 µM at the hα7) comparable to the ureidic derivative PNU-120596, and shows a typical type II PAMs profile with a massive potentiation of agonist-evoked responses and considerable reduction in the rate of desensitization [70]. A-867744 was able to potentiate ACh-evoked currents with an EC_50_ of ~1 µM but with much higher efficacy respect to that for TQS [71].

PNU-120596 and A-867744 have been shown to bind to the same binding site and compete each other. In fact, the presence of A-867744 bound to the receptor slows down the modulation mediated by PNU-120596 by approximately 50 fold, and similarly A-867744 is unable to modulate the receptor if the binding sites are already saturated with PNU-120596 [72]. Through kinetic studies, the most obvious difference between the two compounds is the type of modulation mechanism that they produce. PNU-120596 has a limited ability to bind receptors in the resting state, while A-867744 easily associates to this conformation. Furthermore, PNU-120596 induces radically prolonged channel openings, while A-867744 acts by destabilizing the desensitized state rather than inducing the opening of the receptor. It has been observed that the opening and closing times of the channel cannot exceed 1 ms. Consequently, they act by increasing the mean open time, and not by forming an energetically stable open conformation. A-867744 therefore behaves both as a type II PAM in the prolonged and moderate presence of the agonist, and as a type I PAM after short agonist release pulses. Consequently, it has been shown that specific compounds classified as type II PAMs can behave in different ways. These different mechanisms of action can allow the attainment of derivatives with advantageous properties from a therapeutic point of view. For instance, A-867744-mediated modulation is fast, and has augmented onset and fast offset [72]. 

Other derivatives carrying a tetrahydro-cyclopenta[*c*]quinoline-8-sulfonamide (TQS) nucleus have been synthesized. They are structurally characterized by the presence of a phenyl ring carrying from 1 to 5 methyl groups in various positions, and by a different stereochemistry on the cyclopentaquinoline nucleus leading to the *cis*-*cis* or *cis*-*trans* isomers shown in Figure 10. The number and position of methyl substitution on the phenyl ring and the different stereochemical arrangement gave compounds with diverse pharmacological profiles. *Cis*-*cis* isomers **1** can be allosteric agonists or type II PAMs, while *cis*-*trans* isomers **2** can behave as type I PAMs, NAMs or SAMs [73]. The importance of the *cis-cis* isomerism in the activity as allosteric agonists is also confirmed by the derivative GAT-107 which is the *cis-cis*-(+) enantiomer of 4BP-TQS and which in fact acts as an allosteric agonist. Allosteric modulators are particularly sensitive to minimal changes in the chemical structure giving extremely different pharmacological profiles. “Knife-edge” structure–activity relationship is a characteristic of the allosteric ligands due to the fact that minimal chemical modifications can cause a change in activity from NAMs or SAMs to PAMs and vice versa [42]. This fact can be exploited by medicinal chemistry to achieve large pharmacological differences through minimal chemical changes.

Optical isomerism can also affect the pharmacological profile of the modulators. Racemic 2,3,5,6TMP-TQS (Figure 10) was initially identified as SAM, since it has a silent behavior on the activation by 100 µM ACh interacting with an allosteric site, and as allosteric antagonist since it was able to reduce the activation by GAT-107 (structure reported in Figure 2) [16]. A deeper investigation on the activity of this racemate also showed an unexpected effect, the potentiation of the response to low doses of ACh. Indeed, the resolution of the racemic mixture demonstrated that the two enantiomers of 2,3,5,6TMP-TQS have a different activity: the (+) enantiomer behaves as a PAM, while the (−) enantiomer is an allosteric antagonist [74]. 

The arylsulfonamide group is a common feature of several molecules that exhibit an α7-PAM profile, such as the already mentioned A-867744 and TQS (Figure 9), and TBS-516 (EC_50_ = 0.4 μM at the hα7) (Figure 11). These three compounds differ in the core structure (pyrrole, cyclopenta[*c*]quinoline, or triazole) and in the organization of the groups on it. They are supposed to bind to the same intersubunit site, and thus their interactions are mutually exclusive. Docking studies have proposed a different orientation for A-867744 with respect to TQS and TBS-516, suggesting an explanation for its different functional activity on mutated receptors and pharmacological profile. A-867744, but not TQS and TBS-516, is able to displace the binding of α7-selective agonists interacting with the orthosteric site. This effect may be due to conformational changes in the receptor that are different from those caused by other PAMs [75].

Compound **3** [4-(5-(4-chlorophenyl)-4-methyl-2-propionylthiophen-3-yl) benzenesulfonamide], a structural derivative of A-867744 (Figure 11), is a potent (EC_50_ = 0.32 μM at the hα7) and selective α7-PAM. Phase 1 clinical trial indicated that compound **3** has a safe profile and excellent brain penetration. Furthermore, it demonstrates excellent efficacy in improving short-term episodic and working memory deficits in preclinical models of cognitive dysfunction. It is under development as monotherapy for mild and moderate Alzheimer’s disease [28].

A close structural analog, LL-00066471 (a type II PAM), shows high potency (EC_50_ = 0.49 μM at the hα7), and good brain permeability with a brain:plasma ratio around 1.5 in rats. LL-00066471 demonstrated high efficacy in animal models of cognitive and sensorimotor-gating deficits, supporting its potential use in the treatment of cognitive impairments associated with neurological and psychiatric conditions [76,77]. At Gedeon Richter Plc., A-867744 has been extensively modified to discover novel α7-PAMs. Among the synthesized molecules, pyrazole **4** has proved to be particularly interesting. It shows a significant potency (EC_50_ = 1.2 μM at the hα7) and efficacy for the α7 receptor and ability to improve cognitive performance in vivo in rodent models [78].

Through the screening of libraries of natural small molecules whose neuroprotective, anti-inflammatory, and pro-cognitive effects were known, various flavonoids were isolated, including the isoflavonoid genistein (EC_50_ = 9.36 μM at the hα7), the flavonol quercetin (EC_50_ = 15.39 μM at the hα7), and the neoflavonoid 5,7-dihydroxy-4-phenylcoumarin (EC_50_ = 9.64 μM at the hα7) (Figure 12). Although these natural compounds behave as type I α7-PAMs, they interact with an allosteric site located in the archetypical TMD site [79], suggesting that the interaction with the TMD site might induce type I- or type II-like effects depending on the respective affinities and transduction process at this site.

Chalcone-like structures such as compounds **5** (EC_50_ = 25.0 μM) and **6** (EC_50_ = 3.3 μM), were obtained from isoliquiritigenin (EC_50_ = 11.6 μM), a major constituent of the *Glycyrrhizae rhizoma*, which showed specific activity as PAMs of the hα7. Of all the derivatives obtained through modifications in the number and position of phenolic functions, derivatives **5** and **6** proved to be the most interesting since, despite the minimal differences, derivative **5** behaves as a type I PAM, while derivative **6** behaves as a type II PAM [80]. The OCH_3_-substituted derivative **7** (IC_50_ = 9.1 μM at the hα7) acts as an α7-NAM, even if a higher inhibition was obtained with the saturated analogue **8** (IC_50_ = 2.1 μM at the hα7) [81]. The fluoro-substituted RGM079 (EC_50_ = 8.3 μM) of the polyhydroxy chalcones behaves as a potent PAM at the hα7 and shows neuroprotective properties in Alzheimer’s disease toxicity models [82]. 

Curcumin (EC_50_ = 0.21 μM at the hα7), which is the most abundant polyphenolic component extracted from turmeric, the rhizome of *Curcuma longa*, has long been used in traditional medicine as an anti-inflammatory, anticancer, antioxidant, and antimicrobial. Other beneficial effects have also been shown in animal model of Alzheimer’s and Parkinson’s disease, but the mechanism of action is not known [83]. Several phase 2 clinical trials of curcumin for mild cognitive impairment and mild to moderate Alzheimer’s disease have been completed, but results are unknown. Recently, the effects of curcumin on human α7 receptors have been investigated and curcumin has been shown to act as a PAM. It markedly decreases receptor desensitization and reactivates desensitized receptors, thus acting as type II PAM. This mechanism of action could therefore be responsible for the analgesic and anti-inflammatory actions of this compound [84].

The α7-PAMs JNJ39393406 and AVL-3288, and the α7 silent agonist ASM-024 (Figure 13) showed good preclinical profiles but unfortunately JNJ39393406 and AVL-3288 did not show efficacy in clinical trials. JNJ39393406 has completed two phase II trials for smoking cessation and another trial to treat cognitive or depressive symptoms in patients with unipolar depression [85,86], while the trials for schizophrenia and Alzheimer’s disease were terminated. AVL-3288 has completed a phase I trial for the treatment of schizophrenia and schizoaffective disorders [87]. The clinical results showed that although this compound was well tolerated at different doses, it improved neither the primary target engagement (P50) nor cognitive outcomes in schizophrenia, indicating lack of efficacy for AVL-3288. ASM-024 has completed phase II trials for the treatment of chronic obstructive pulmonary disease, mild allergic asthma, and asthma, respectively [88]. The results indicated that this PAM improves bronchial airway hyperreactivity and asthma induced by methacholine but not induced by allergens.

## 9. Positive Allosteric Modulators of the α4β2 nAChR

The heteromeric α4β2 nAChR subtype constitutes approximately 10–37% of the total nAChRs, depending on the various brain regions [89]. α4β2 nAChRs can have different stoichiometries with distinct functional properties and different sensitivity towards agonists: the low-sensitive (LS) (α4)_3_(β2)_2_ stoichiometry and the high-sensitive (HS) (α4)_2_(β2)_3_ stoichiometry. The HS subtype is most strongly affected by chronic nicotine exposure, causing a marked nicotine-induced potentiation, in the brain of rodents. Post-mortem studies on smokers’ brains have confirmed an increased expression of this receptor [90]. Moreover, HS is the most sensitive to desensitization induced by low agonist concentrations, although the LS stoichiometry has a faster desensitization kinetics than the HS receptor [91]. 

TC-5214 ((*S*)-(+)-mecamylamine) is both a PAM modulator of the HS stoichiometry and a selective and more efficacious blocker of LS stoichiometry, than TC-5213 ((*R*)-(−)-mecamylamine) [92]. Mecamylamine isomers differ in the interaction at binding sites in the TMD which could explain the different pharmacological profile of each isomer at each stoichiometry [93]. Given the complexity of the α4β2 subtype, in recent years medicinal chemistry has increased attention on the stoichiometry-selective allosteric modulation of this receptor even if the number of α4β2 allosteric activators is still rather limited [94]. 

*des*-Formylflustrabromine (dFBr) (Figure 14), one of the 22 alkaloids isolated from the marine organism *Flustra foliacea*, is the first PAM with high selectivity for the α4β2 nAChR subtype, potentiating the action of ACh and other agonists [95]. dFBr increases cognition in both naïve and cognitively impaired rats, supporting the strategy of using α4β2 PAMs for the treatment of cognitive deficits [96]. dFBr has been used as a unique lead for structure–activity relationship studies to understand what structural features are important for potentiation and/or inhibition of the ACh activity.

Various cations, such as zinc (Zn^2+^) and calcium (Ca^2+^) [97,98], seem to play crucial roles in the modulation of various nAChR subtypes in the brain. Co-application of Zn^2+^ (50 µM) with a saturating ACh concentration was shown to potentiate ACh-evoked currents by 260% [99]. It was also observed that Zn^2+^ at high concentrations inhibits (IC_50_ = 440 µM) different nAChR subtypes (i.e., α2β2, α2β4, α3β4, α4β2, and α4β4), while at low concentrations it potentiates (EC_50_ = 16 µM) the same subunit combinations [99]. In any case, Zn^2+^ does not affect the desensitization rate of nAChRs regardless of the combination of subunits. The Zn^2+^ cation can produce positive and negative modulation on the α4β2 nAChR by binding to two different domains. Furthermore, the allosteric modulation mediated by Zn^2+^ depends on the receptor stoichiometry: the allosteric inhibition on both receptor stoichiometries is due to the coordination of zinc at sites in the β2(+)/α4(−) interface, while if the stoichiometry is 3:2 the coordination of zinc in the α4(+)/α4(−) interface induces allosteric potentiation [97,100]. 

NS-9283, discovered by NeuroSearch A/S, modulates only the α4β2 LS stoichiometry (3:2) (EC_50_ = 0.11 μM), and the binding site was shown to be associated with the α4(+)/α4(−) interface only present in this stoichiometry [101]. NS-9283 acts as a PAM on (α2)_3_(β2)_2,_ (α2)_3_(β4)_2,_ (α4)_3_(β2)_2_, and (α4)_3_(β4)_2_ subtypes with relatively equivalent potency. On receptors containing the α3 subunit as well as in the stoichiometric combination 2α:3β, NS-9283 has no effect [102]. Therefore, this different receptor stoichiometry can be fundamental for having selective PAMs as evidenced for Zn^2+^ and NS-9283 which are completely selective towards the LS (α4)_3_(β2)_2_ receptors. Unlike NS-9283, NS-206 modulates both 2α:3β and 3α:2β stoichiometries of the α4β2 nAChR (EC_50_ = 2.2 and 0.16 μM, respectively). Interestingly, the modulatory activity is due to the α4 subunit, while the β-subunit has no influence. In fact, the same PAM activity is present on both α4β2 and α4β4 combinations [101]. 

HEPES [4-(2-hydroxyethyl)-1-piperazineethanesulfonic acid] is able to discriminate between LS and HS α4β2 receptors. More specifically, HEPES potentiates HS α4β2 receptors (EC_50_ = 5.7 μM) in the µM concentration range with little effect on LS α4β2 receptors (EC_50_ = 54.0 μM). As previously stated, the HS (α4)_2_(β2)_3_ stoichiometry has a β2(+)/β2(−) interface which is not present in the LS stoichiometry. Since mutagenesis studies showed that the mutation on β2-D218, at the β(+) face, abolishes HEPES potentiation, the allosteric site was located precisely at the β2(+)/β2(−) interface of the HS α4β2 subtype [103]. 

CMPI was discovered by Amgen Inc. through modifications starting from the piperidinamide **9**, which had been identified through a high-throughput screening. To increase efficacy and potency, Albrecht and coworkers inserted bioisosters of the amide group such as pyrazole, thiazole, or imidazole. A series of interesting derivatives acting as potent modulators of the rat and human α4β2 receptors were thus obtained, capable of penetrating the CNS in an excellent manner [104]. CMPI was able to discriminate the two stoichiometric combinations of α4β2 receptors. More precisely, CMPI positively modulated LS α4β2 (3:2) but not HS α4β2 (2:3) nAChRs [105,106]. Amgen Inc. also discovered carbamates as α4β2 receptor enhancers starting from a lead compound (**10**) with an urea function and a stereogenic center. The (*R*) enantiomer was a moderate potentiator with selectivity for hα4β2 over hα3β2 and hα3β4. Potency and efficacy were improved transforming the urea into carbamate, and changing the substituents on the phenyl ring to give **11**, which is an enhancer of the α4β2 receptor [107].

Regarding the allosteric modulatory sites at the α4β2 nAChR [49,51], the PAM site for galantamine, medication used in the treatment of Alzheimer’s disease at doses lower than those for acetylcholinesterase inhibition [108], has been located in the ECD within the α subunit at ~12 Å from the agonist site [109,110]. The site for Zn^2+^ is located on the ECD, involving non-canonical residues within the β2(+)/α4(−) or α4(+)/α4(−) interface based on α4β2 stoichiometry as previously reported [111]. Neuronal AChRs have two zinc binding sites located at non-canonical subunit interfaces alternated with ACh sites at canonical subunit interfaces. Allosteric sites are also present in the cytoplasmatic domain for α4β2 nAChRs. The cytoplasmatic or intracellular domain (ICD) has several phosphorylation sites that can modulate receptor functions through intracellular signaling pathways [112].

Given the low number of identified α4β2 PAMs, much work still needs to be performed to understand the action that modulators have on receptors with different stoichiometry. α4β2 nAChRs have been studied as drug targets in different therapeutic areas [2] and the potential of α4β2 PAMs strongly encourages the exploration of these derivatives as potential treatments of different CNS pathologies.

## 10. Allosteric Modulators at Other Non-α7 nAChR Subtypes

AN6001 is the first allosteric modulator that has been identified and characterized (Figure 15) with selectivity for α6β2-containing nAChRs (i.e., α6β2*) (EC_50_ = 0.58 μM), and no modulatory activity on α4β2, α7, α3β4, and muscle-type receptors [113]. AN6001 increased the potency and efficacy at the human recombinant α6/β2β3, α6/α3β2β3 and a concatenated version of human α6/α3β2β3 assembled as β3–α6–β2–α6–β2. In addition, AN6001 increased agonist-induced dopamine release from striatal synaptosomes. Ongoing studies on the expression and function of α6β2* nAChRs in the brain will provide information not only on these receptors, but also on the potential use of ligands such as AN6001 as pharmacological tools for neurological conditions involving this receptor subtype.

Morantel (Figure 15) is a widely used anthelmintic, not for human use, in worm infections, but already described as a PAM of hα3β2 nAChRs (EC_50_ = 60 μM) [114]. Morantel increases the current when applied together with ACh by binding to non-canonical sites in the β(+)/α(−) subunit interface [115,116]. 

α9α10 nAChRs expressed in cochlear hair cells consist exclusively of α9 and α10 subunits with different stoichiometries (α9)_3_(α10)_2_ and (α9)_2_(α10)_3_ [117]. The knowledge regarding the number of binding sites in the different stoichiometries as well as the contribution of each subunit to the binding sites is not fully known yet, even considering that the crystal structure of the ECD of the homomeric α9 subunit is available [118]. The pharmacology of α9α10 nAChRs is different compared to other nAChR subtypes. In addition to be inhibited by the classical competitive antagonist α-bungarotoxin, α9α10 nAChRs are blocked by nicotine [119], the GABA_A_ receptor antagonist bicuculline, the glycine receptor antagonist strychnine, and the muscarinic receptor antagonist atropine, whereas the muscarinic agonist oxotremorine behaves as a partial agonist of this receptor subtype [120]. 

An exhaustive literature search provided evidence that ryanodine and ascorbic acid are modulators of α9α10 nAChRs (Figure 15). More specifically, ryanodine is an alkaloid obtained from the *Ryania speciosa* plant, known mostly for its use as an insecticide and for its action in the release of Ca^2+^ through channels known as ryanodine receptors. Ryanodine also modulates the potency and maximal response to ACh, supporting the classification of PAM at α9α10 nAChRs [121]. Ascorbic acid selectively potentiates ACh-evoked responses at α9α10 nAChRs, with no apparent effect at α4β2 and α7 nAChRs. Since ascorbic acid also has intrinsic partial agonist activity at α9α10 nAChRs, it has been hypothesized that it binds to an allosteric site that triggers receptor activation, and thus, behaving as an ago-PAM [122].

## 11. Negative Allosteric Modulators of nAChRs

Negative allosteric modulators (NAMs) inhibit agonist-evoked activity by inducing conformational changes in the receptor that decrease the probability of channel opening. Dehydronorketamine (DHNK) (Figure 16) is a NAM with higher selectivity for α7 over α3β4 nAChRs [123]. DHNK is a metabolite of the general anesthetic (*R*) (*S*)-ketamine, and the fast-acting antidepressant esketamine (i.e., (*S*)-ketamine). DHNK was able to prevent the increase in ethanol consumption induced by alcohol exposure during adolescence in rat [124].

1,2,3,3a,4,8b-haxahydro-2-benzyl-6-*N,N*-dimethylamino-1-methylindeno[1,2-b]pyrrole (HDMP), obtained by a conformationally restricted analogs of nicotine [125], shows 350-fold higher inhibitory potency on α7 than on α3β4 and α4β2. HDMP is thus an example of NAM with a good selectivity profile towards the α7 nAChRs. Phencyclidine (PCP) is an α7 NAM 18-fold less potent than HDMP with a different selectivity profile. In fact, PCP inhibits the α7 nAChR with potency 23-fold higher than that at the α4β2 nAChR, but similar as that for the α3β4 nAChR [126].

Several derivatives identified as NAMs are described in the literature which show a certain selectivity for the human α4β2 nAChRs with respect to hα3β4 nAChRs (Figure 16). These molecules were identified and characterized some years ago while no new compounds have been recently reported in the literature. Researchers at Ohio University discovered KAB-18 [127], a piperidine derivative showing selectivity for the hα4β2 over the hα3β4 subtype in recombinant systems [128]. The screening of a set of analogues (some are reported in Figure 16) allowed to derive some structure–activity relationships. The presence of the 2-biphenylcarboxylate ester group and 3-arylpropyl substituent on the piperidine nitrogen atom of the DDR-18 derivative was required for α4β2 selectivity. The piperidine nitrogen does not need to be basic, as the amide DDR18 has higher potency and similar selectivity with respect to KAB-18. Removal of the piperidine *N*-substituent, as in COB-1, abolished α4β2 selectivity. The same happen if the distal phenyl ring of the biphenyl moiety is replaced by amide or imide, as in IB-2. Studies to discriminate the activity of KAB-18 enantiomers showed similar potency and selectivity for the isomers.

The binding site of NAMs have been studied on α4β2 receptors. Homology modeling, blind docking, molecular dynamics studies, and targeted mutagenesis studies have determined the location of the allosteric binding site for NAMs at hα4β2 receptors. More precisely, the site was located in the interface between subunits α and β 10 Å apart from the orthosteric site [128,129,130]. Since most residues comprising the allosteric site are located on the β2 subunit, the site has been called “β subunit site”. Residues such as Thr58, Phe118, and Glu60, located on the β2 subunit, show <26% homology with other nAChR subunits. Phe118 and Thr58 are not present in hα4β2 and hα3β4. This low conservation could be responsible for the selectivity shown by some NAMs towards different subunits [131].

Virtual screening was applied by the same authors to identify new classes of compounds, exemplified by **12** [132] and **13** [131]. Compound **12**, carrying a sulfonylpiperazine scaffold which is unique among NAMs previously described, shows approximately 12-fold selectivity for hα4β2 nAChR over the hα3β4 subtype. Although the potency of **13** is in the same range as that for **12**, the selectivity for the hα4β2 subtype is 5-fold lower.

From the screening of a library of GABA_A_ allosteric modulators, UCI-30002 was identified as a non-selective NAM of neuronal nAChRs. When tested in vivo, UCI-30002 blocks nicotine self-administration in rats at brain concentrations consistent with nAChR inhibition.

## 12. Concluding Remarks

Previous and current studies support the view that α7 and non-α7 nAChRs are involved in many physiological functions, and consequently, they are important targets in a variety of pathological conditions. Allosteric modulators have become the most interesting compounds for the development of new medications. Although there are no modulators in the market yet, several prospective compounds have demonstrated beneficial effects in clinical trial II (e.g., ASM-024), increasing the possibility of being approved for chronic obstructive pulmonary disease and asthma. We expect new promising candidates for a variety of neurological uses in the near future.

In addition to the obvious clinical interest, allosteric modulators can give us a different structural and functional perspective of how receptors are activated and desensitized compared to agonists. For example, if the main sites for type I and type II PAMs are, respectively, located in the ECD and TMD, what is the transduction mechanism involved in each interaction, and how agonist binding to the orthosteric site trigger these two different mechanisms?

It is very clear that different molecular structures can allosterically modulate distinct nAChR subtypes, and that small structural modifications induce important changes in the pharmacological profile of the studied compounds. The observed structural and functional flexibility of modulators will hopefully allow the development of novel compounds with improved receptor selectivity and efficacy for specific clinical uses, including neurological and non-neurological conditions.

## Figures and Tables

**Figure 1 molecules-28-01270-f001:**
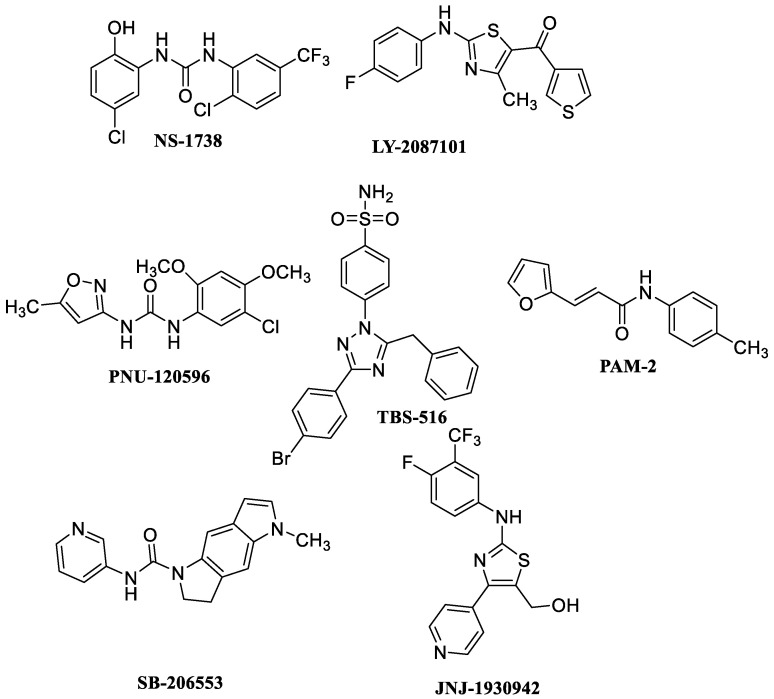
Examples of type I, type II and type I/type II intermediate α7-positive allosteric modulators (PAMs).

**Figure 2 molecules-28-01270-f002:**
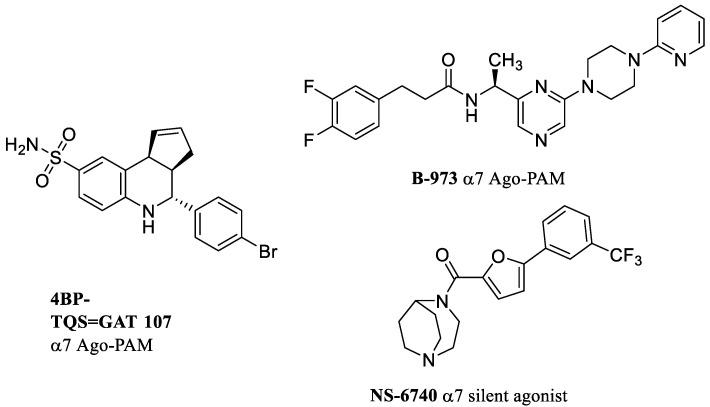
Examples of allosteric activating PAMs (Ago-PAM) and silent agonist.

**Figure 3 molecules-28-01270-f003:**
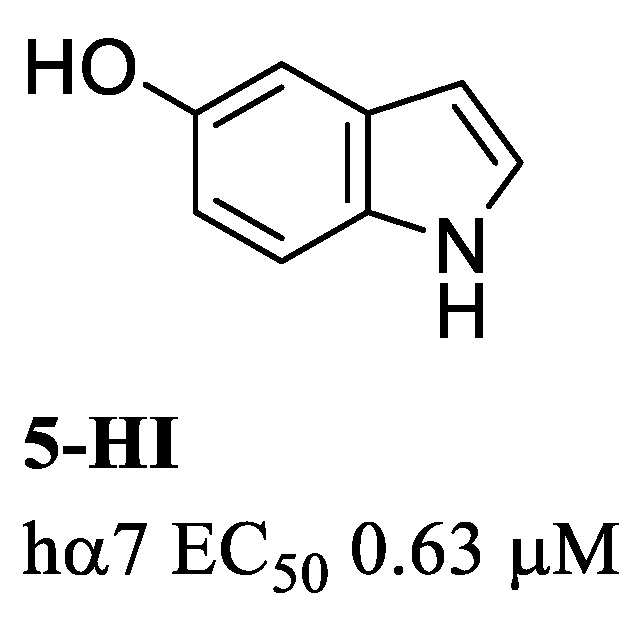
Structure of the first type I PAM discovered.

**Figure 4 molecules-28-01270-f004:**
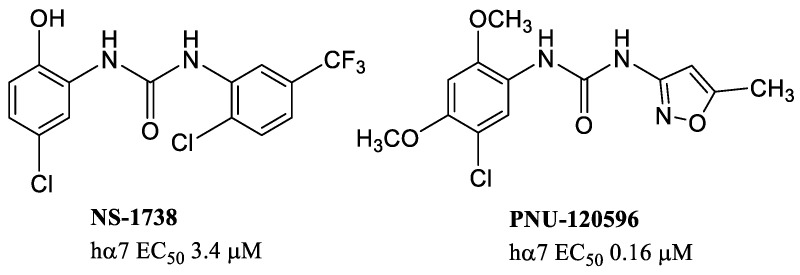
Allosteric modulators with an ureidic function.

**Figure 5 molecules-28-01270-f005:**
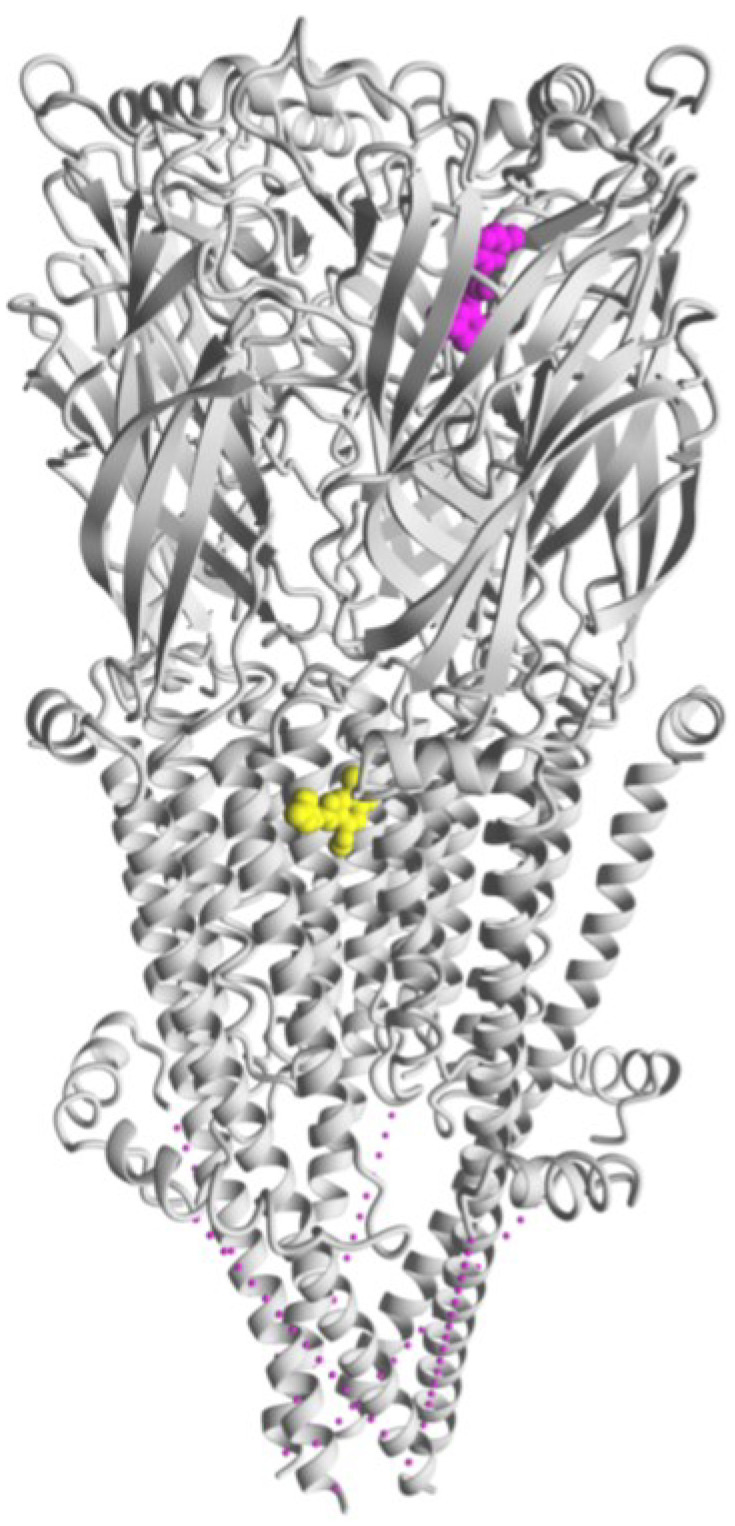
Cryo-EM structure of the human α7 nAChR (pdb:7EKT) showing the intersubunit site for PNU-120596 (yellow), the archetype of type II PAMs, located in the TMD (modified from reference [58]). The ECD binding site for NS-1738 (magenta), a typical type I PAM, was obtained by molecular docking as previously described [31]. Only one of five sites per each PAM is shown for simplicity.

**Figure 6 molecules-28-01270-f006:**
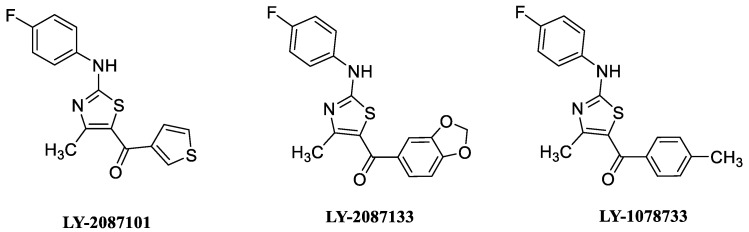
Examples of non-selective αAChR PAMs.

**Figure 7 molecules-28-01270-f007:**
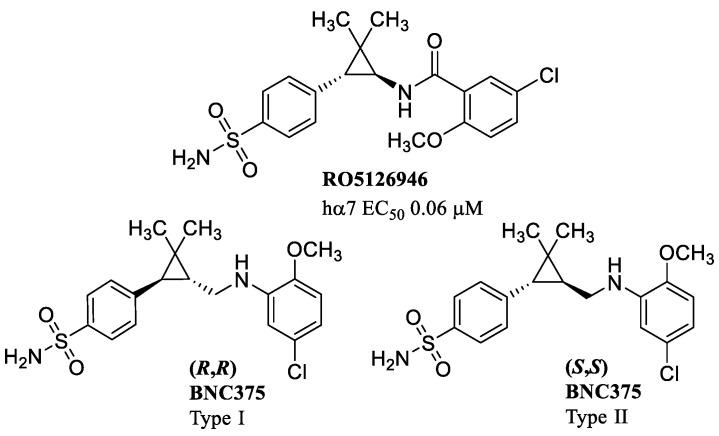
Examples of type I and type II α7-PAMs with a cyclopropane ring.

**Figure 8 molecules-28-01270-f008:**
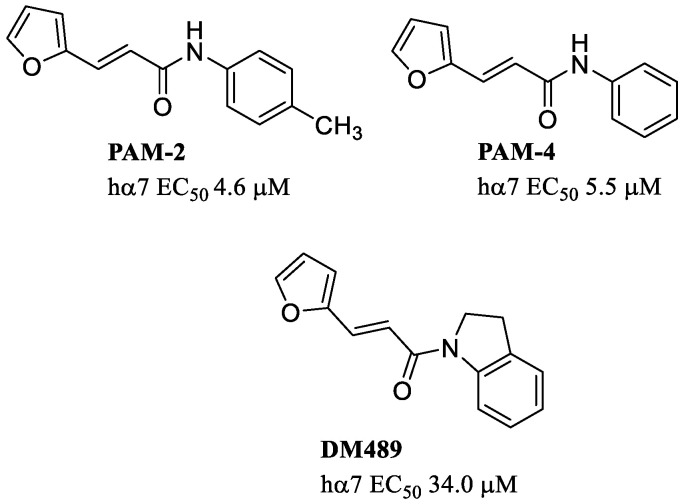
Examples of PAMs with an amide function.

**Figure 9 molecules-28-01270-f009:**
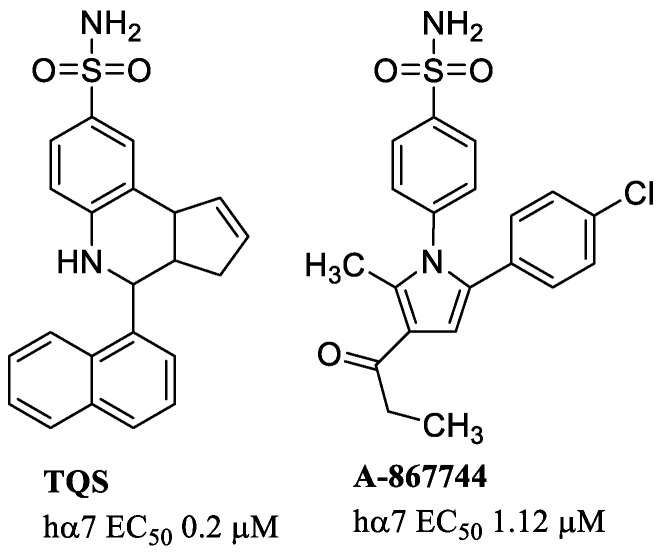
Examples of type II α7-PAMs with arylsulfonamide ring.

**Figure 10 molecules-28-01270-f010:**
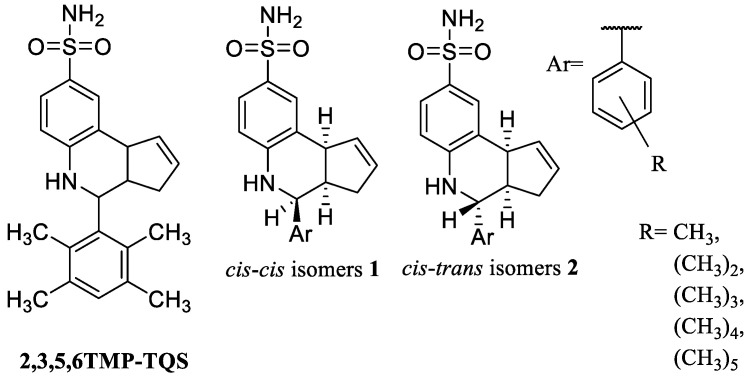
Allosteric modulators with different pharmacological profile based on stereochemistry.

**Figure 11 molecules-28-01270-f011:**
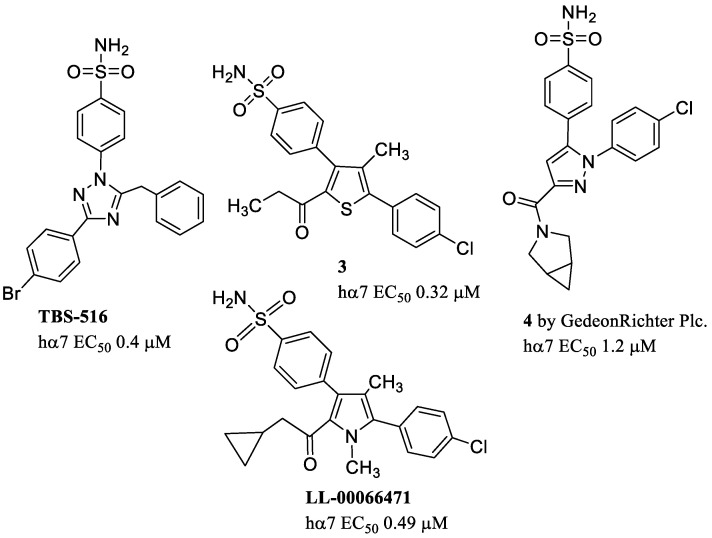
Arylsulfonamide allosteric modulators.

**Figure 12 molecules-28-01270-f012:**
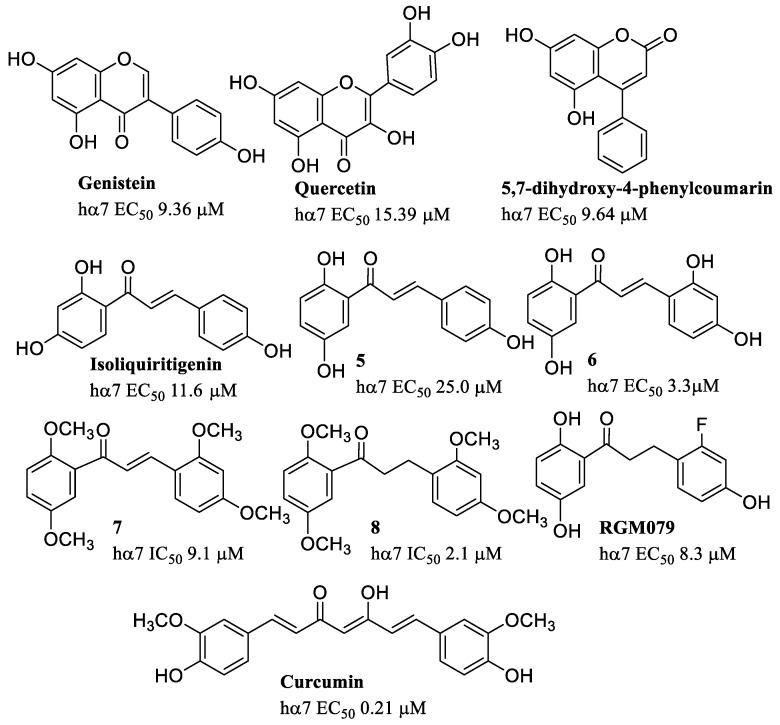
α7-PAMs from natural sources.

**Figure 13 molecules-28-01270-f013:**
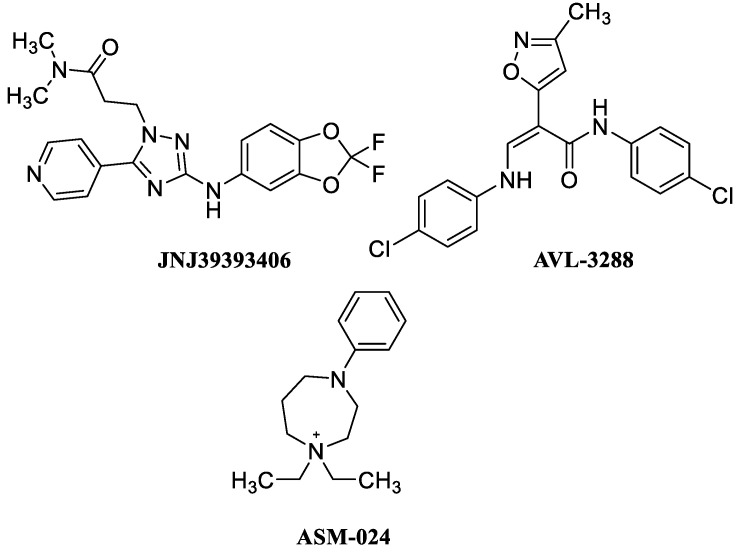
Nicotinic agents in clinical trials by U.S. National Institutes of Health.

**Figure 14 molecules-28-01270-f014:**
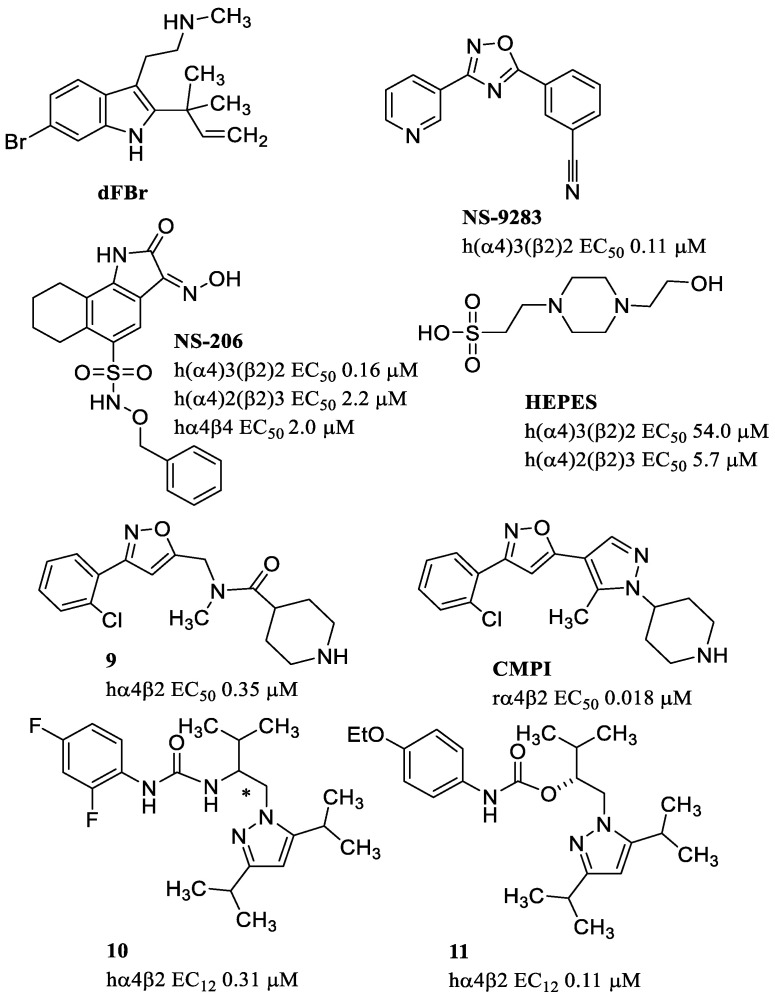
Allosteric modulators of the α4β2 nAChR.

**Figure 15 molecules-28-01270-f015:**
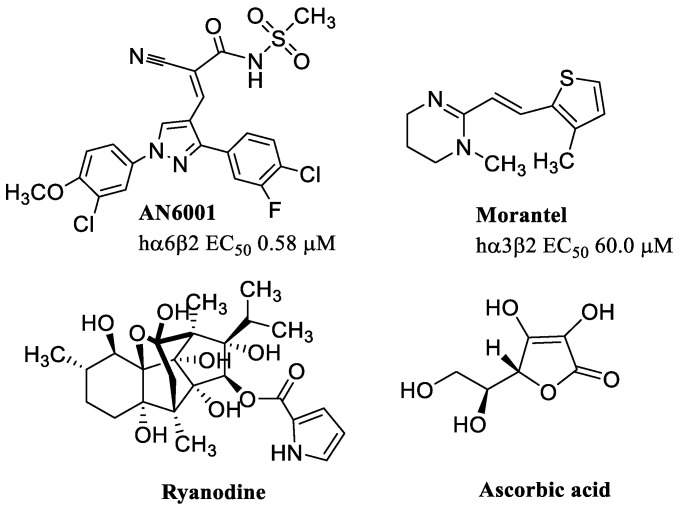
Allosteric modulator of α6β2*, α3β2, and α9α10 nAChR subtypes.

**Figure 16 molecules-28-01270-f016:**
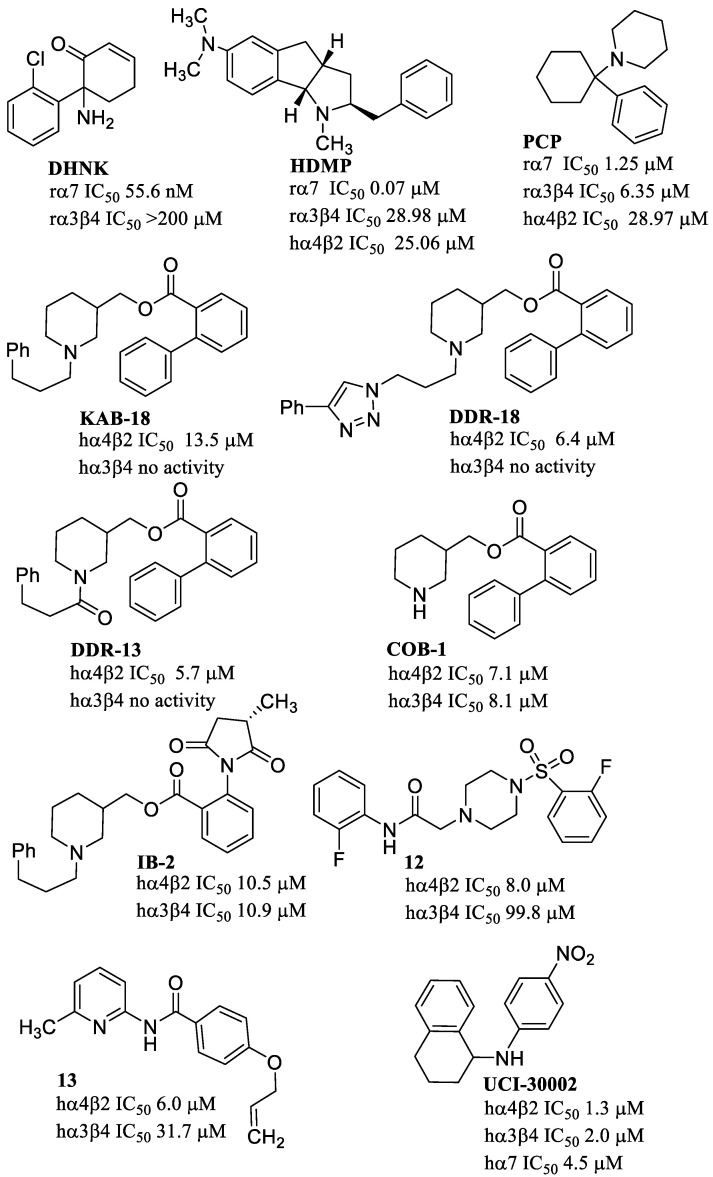
Negative allosteric modulators (NAMs) of different nAChR subtypes.

## Data Availability

Not applicable.

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
