# Peer review of "Recent Advances in the Discovery of Nicotinic Acetylcholine Receptor Allosteric Modulators"

_molecules, 2023, doi:10.3390/molecules28031270_

Round 1

Reviewer 1 Report

The manuscript needs a section detailing how the selection of bibliography was conducted: which years were included, what keywords were used, what were the criteria for inclusion/exclusion, etc.

Line 112: Provide the meaning of a7(+)/a7(-) for readers less knowledgable in the field.

Line 246-247: the claim regarding natural products libraries requires a reference.

Lines 270-271: Explain the reason why PNU-120596 was not approved by the FDA according to the information available. Perhaps toxicity?

288-289: Provide exemples for the pathologies mentioned.

Line 352: Probability is not the best term to describe the opening of the channel, please rewrite.

Lines 372-374: The idea conveyed by these lines had been mentioned earlier in the text, consider deleting.

Line 379: I disagree that the effect descibed is “obvious”. In fact, there are several exemples in which optical isomerism does not affect the pharmacological effect significantly.

There are quite some missconceptions regarding the phytochemistry of some natural molecules mentioned. For exemple “genistein with an isoflavonoid group” is not correct, as genistein is itself an isoflavonoid, a subclass of flavonoids. The same aplies to “quercetin with a flavonol group”, as quercetin is itself a flavonol.

Line 431: Refer to “A chalcone”, instead of “flavonoids with a chalcone structure”.

Line 445-446: Claims in these lines must be better substanciated or removed. Most effects of curcumin found in animal experiments have not been reproduced in humans. If these claims refer to animal models, clearly state so.

Lines 454-460: What are the results of the phase I and phase II trials cited using [84] and [85]? Include in the text.

Line 488: the clinical trial cited seems to address a diferent molecule than the one in Figure 14. Although they are similar, they are not the same. If this is the case, remove the mention to this trial.

The overall section division of the work must be revised. For exemple, it makes no sense to have a sectio 10 if it includes only one paragraph. Furthermore, there is a already a section for non-a7 subtypes (section 11), so what is the rationale for this division?

Concluding remarks must be revised. Culprit (line 650) is not an adequate word in this context and 650-651 must be revised as it does not seem to be adequately constructed. Line 651 seem to formulate a question, however there is no question mark.

Author contributions are not aligned with the journal’s guidelines. Authors should state the specific role of each co-author to the multitude of diferente tasks associated to this work.

Reviewer 2 Report

The review article authored by Dina Manetti et al is a timely update on the advances in the very important topic of allosteric modulation of nicotinic acetylcholine receptors. Overall it is very well organized and written and should have a reasonably wide audience. I recommend the manuscript be published. However I do have a few suggestions.   1. In the introduction part, I suggest the authors include a brief plan on how they will be presenting the review to the readers.   2. In the beginning and end of the numbered sections, I suggest the addition of an overview and summary sentence respectively.   3. I suggest the inclusion of  the following review article. Roger L. Papke and Nicole A. Horenstein
Therapeutic Targeting of a7 Nicotinic Acetylcholine Receptors
Pharmacol Rev 73:1118–1149, July 2021   4. There are a few grammatical errors.  Line 21 "are therefore an opportunity" Line 651 "when do not work properly" Line 656 "We expect have" Line 663 Question mark?
